# Safe-Shields: Basal and Anti-UV Protection of Human Keratinocytes by Redox-Active Cerium Oxide Nanoparticles Prevents UVB-Induced Mutagenesis

**DOI:** 10.3390/antiox12030757

**Published:** 2023-03-20

**Authors:** Francesca Corsi, Erika Di Meo, Daniela Lulli, Greta Deidda Tarquini, Francesco Capradossi, Emanuele Bruni, Andrea Pelliccia, Enrico Traversa, Elena Dellambra, Cristina Maria Failla, Lina Ghibelli

**Affiliations:** 1Department of Biology, University of Rome Tor Vergata, 00133 Rome, Italy; 2Department of Chemical Science and Technologies, University of Rome Tor Vergata, 00133 Rome, Italy; 3Experimental Immunology Laboratory, IDI-IRCCS, 00167 Rome, Italy; 4Molecular and Cell Biology Laboratory, IDI-IRCCS, 00167 Rome, Italy

**Keywords:** cerium oxide nanoparticles, SOD and catalase mimetic, keratinocytes, UV exposure, DNA damage, mutagenesis, UV protection

## Abstract

Cerium oxide nanoparticles (nanoceria), biocompatible multifunctional nanozymes exerting unique biomimetic activities, mimic superoxide-dismutase and catalase through a self-regenerating, energy-free redox cycle driven by Ce^3+/4+^ valence switch. Additional redox-independent UV-filter properties render nanoceria ideal multitask solar screens, shielding from UV exposure, simultaneously protecting tissues from UV-oxidative damage. Here, we report that nanoceria favour basal proliferation of primary normal keratinocytes, and protects them from UVB-induced DNA damage, mutagenesis, and apoptosis, minimizing cell loss and accelerating recovery with flawless cells. Similar cell-protective effects were found on irradiated noncancerous, but immortalized, p53-null HaCaT keratinocytes, with the notable exception that here, nanoceria do not accelerate basal HaCaT proliferation. Notably, nanoceria protect HaCaT from oxidative stress induced by irradiated titanium dioxide nanoparticles, a major active principle of commercial UV-shielding lotions, thus neutralizing their most critical side effects. The intriguing combination of nanoceria multiple beneficial properties opens the way for smart and safer containment measures of UV-induced skin damage and carcinogenesis.

## 1. Introduction

Nanozymes are emerging catalytic tools exploiting the similar-enzymatic properties certain inorganic materials acquire when in the nanoscale [1]. Nanozymes are extensively used in industrial catalysis [2]. While the specific activity of protein enzymes relies on molecular complexity, nanozymes are amazingly simple, consisting of atom arrays of just one or two elements [3]. The reactive surface atoms of nanozymes behave as the metal atoms present as prosthetic groups in the active site of biological enzymes. However, if the protein fold is the context favouring catalysis in biological enzymes, in nanoparticles this is due to nanoparticle lattice, which also provides the conditions for recovering the original atomic state, thus rendering the nanozyme reactivity essentially self-regenerating and energy free [4].

Nanoceria, whose lattice is made of Ce and O atoms, are a unique example of multifunctional nanozymes, performing an unrelated series of similar-enzyme functions on biomolecules, including dehydratase, phosphatase, the antioxidant superoxide dismutase (SOD), catalase, and glutathione peroxidase [5,6,7,8,9,10]. The presence of Ce^3+^ atoms on nanoceria surface, which is limited to nanoscale ceria, allows the redox catalytic activities. SOD and catalase-like functions, which have been thoroughly studied [11,12], occur as coupled events, consisting in the redox switch between the two valence states of Ce ^(3+/4+)^, which get cyclically oxidized while reducing superoxides to peroxides (SOD-mimesis), and reduced back while oxidizing peroxide to molecular O_2_ (catalase mimesis), thus completing the cycle returning to the original valence state without altering the crystalline structure of the nanoparticle. Nanoceria redox cycle thus allows scavenging the most noxious biological reactive oxygen species (ROS) in a self-regenerating, energy-free series of events, being antioxidants with unprecedented pharmacological potential [13]. Ce^3+^ atoms, responsible for the redox catalytic activity, are compensated by the corresponding oxygen vacancies, which thus constitute an additional surface defect. It is emerging that also oxygen vacancies possess enzymatic activity of their own [14,15], for example, dehydratase [8]. This suggests that the different catalytic activities on nanoceria surface may take place independently from each other, possibly coexisting and acting simultaneously, to achieve results depending on the combination of their intrinsic properties. To note, nanoceria catalytic activities may be influenced by specific chemical features of the biological environment, such as ionic composition or acidity [16]; for example, a strongly acidic pH (<4) differently affects the catalase vs. SOD-mimetic activities, leading to peroxide accumulation in the acidic (e.g., lysosomic) bioenvironment [10,17]. The high biocompatibility of nanoceria emerging from in vitro and in vivo nanotoxicology studies [18] strongly suggests multiple fruitful, ad hoc pharmacological exploitations of these intriguing properties.

Nanoceria possess the additional non-enzymatic function of absorbing UV photons, in a similar guise as do titanium dioxide nanoparticles (nanotitania and TNPs), a main component of modern commercial sun-shield lotions. [19,20]. Especially in the anatase form, nanotitania are highly efficient UV shields absorbing UV photons [21], thus providing long-lasting protection [20]. This implies production of ROS through a process known as the photocatalytic effect, which allows the dissipation of the extra energy deriving form of UV-photon absorption [22]. This does not cause problems on intact skin, since the stratum corneum constitutes an efficient barrier to both nanoparticles and ROS. However, in the case of compromised stratum corneum (e.g., erythematous or wounded skin), nanotitania may penetrate the skin and, upon irradiation, oxidatively damage living cells present in the internal epidermis layers. On this issue, we have shown with a principle study that nanotitania are highly toxic when irradiated in contact with cells, strongly increasing cell death and mutagenesis [23].

The diffusion of solar shield cosmeceutical formulations has contributed to the hugely increased habit of leisurely solar exposure, limiting immediate adverse effects such as sunburns and erythema [24]. In spite of this, epidemiology studies covering several decades have shown that commercial solar shields have hardly reduced skin cancer incidence [25,26]. This apparent paradox, which may be in part attributed to individual habits and environmental cues, stresses the urgent need to produce safer and more efficient cosmeceutical solar shields.

We have previously shown, with a proof-of-principle study performed on a reference cell system, that nanoceria efficiently prevent UV-induced cell damage, apoptosis, and mutagenesis via their SOD–catalase mimetic activities; importantly, we showed that nanoceria are even able to scavenge ROS produced by the nanotitania-photocatalytic effect, protecting from irradiated nanotitania-induced apoptosis [23]. Therefore nanoceria, combining antioxidant and UV shielding properties, promise to be an unprecedented multifunctional tool as solar shields.

To verify such an issue on one of the most relevant biological system targets of UV-induced damage, i.e., the skin, we explored the effect of nanoceria on UV-irradiated keratinocytes, using the HaCaT immortalized keratinocyte cell line and primary human keratinocytes.

## 2. Materials and Methods

### 2.1. Nanoparticle Synthesis and Characterization

#### 2.1.1. Cerium Oxide Synthesis

Cerium oxide nanoparticles (CNPs and nanoceria) were synthesized using the sol-gel method. Pluronic F-127 (6.5 g, 0.08 mol) (Sigma-Aldrich, St. Louis, MO, USA) was dissolved in 300 mL of Milli-Q water. After 1 h, Ce(NO_3_)_3_·6H_2_O (15.49 g, 0.036 mol) (Sigma-Aldrich, St. Louis, MO, USA) was poured into the solution, followed by the addition of N, N, NO, NO-tetramethylethylenediamine (TEMED 17 mL, 0.113 mol) (Sigma-Aldrich, St. Louis, MO, USA). The solution was kept overnight under mild stirring. Nanoparticles were then washed in water to remove TEMED traces, collected, and dried overnight at 80 °C.

#### 2.1.2. Titanium Dioxide Nanoparticles Synthesis

Titanium dioxide nanoparticles (TNPs, nanotitania) anatase nanopowders were prepared according to [27,28]. Titanium tetraisopropoxide, Ti(OiPr)_4_, (8 mL, 27 mmol) (Sigma-Aldrich, St. Louis, MO, USA), was dissolved in 92 mL of absolute ethanol, followed by drop addition of a solution of ethanol/water 1:1 (250 mL) under N_2_ flux. The suspension was kept stirring for 10 min, then filtered to obtain a white precipitate, which was dried in air (100–110 °C) for 15 h. The dried precursor was calcinated at 450 °C for 4 h.

#### 2.1.3. Nanoparticles Physicochemical Characterization

The phase and morphology of the nanoparticles were analyzed by X-ray powder diffraction (XRD) analysis using an X-Pert X-Ray diffractometer (Philips PANanalytical, Philips, Amsterdam, Netherlands). The crystal structure was identified by comparison of XRD with the references taken from the JCPDS database (71-1166 for anatase structure and 75-0390 for CNP fluorite structure). The crystallite size of the samples, *d*XRD, was estimated from XRD patterns by measuring the full-width half-maximum (FWHM) of the characteristic peak using the Scherrer equation:
dXRD=0.9 λ /FWHM×cos (θ)
where *λ* is the X-ray wavelength (1.5406 Å) and *θ* is the Bragg angle.

Nanoparticle dimensions were determined using a field emission scanning electron Microscope (FE-SEM) Zeiss Leo Supra 35 (Carl Zeiss SMT, Oberkochen, Germany). Single point-specific surface area measurements were conducted on the powders using BET analysis (TriStar II Plus, ALFATEST, Cernusco sul Naviglio, Italy). Zeta potential and dynamic light scattering (DLS) of the CNPs and TNPs, at a concentration of 200 μg/mL, were measured immediately after sonication at 37 °C in deionized water (pH 7.4) using a Malvern Zetasizer (Nano-ZS, Malvern Instruments, Worcestershire, UK).

### 2.2. Cell Culture

#### 2.2.1. Cells

HaCaT cells (Lonza, Basel, Switzerland), a non-tumorigenic, spontaneously transformed the human keratinocyte cell line [29] and were grown at 37 °C in DMEM medium, supplemented with 10% fetal bovine serum (FBS), 50 mg/L streptomycin, 100,000 units/L penicillin, and 200 mM glutamine (Euroclone, Milan, Italy) in a humidified atmosphere of 5% CO_2_ in the air and routinely split by trypsinization with Trypsin-EDTA (Euroclone, Milan, Italy).

Primary human keratinocytes (KCs) were obtained from skin biopsies of healthy patients undergoing plastic surgery who signed informed consent (IDI-IRCCS Ethical Committee, *n*. 581/3). Briefly, 2 cm^2^ skin biopsies were minced and trypsinized (0.05% trypsin/0.01% EDTA) at 37 °C for 3 h. Cells were collected every 30 min, plated (2 × 10^6^/75 cm^2^ flask) on lethally irradiated 3T3-J2 fibroblast cells, as previously described [30], and cultured in 5% CO_2_ in Rheinwald and Green medium (Dulbecco-Vogt Eagle’s and Ham’s F12 media, 3:1 mixture) containing 10% fetal calf serum (FCS), insulin (5 μg/mL), transferrin (5 μg/mL), adenine (0.18 mM), hydrocortisone (0.4 μg/mL), cholera toxin (0.1 nM), triiodothyronine (20 pM), epidermal growth factor (10 ng/mL), and penicillin/streptomycin (50 IU/mL) (Sigma-Aldrich, St. Louis, MO, USA). Confluent cultures were then trypsinized and cells were plated in secondary cultures or frozen. Second or third passage KCs were used in all experiments and were grown in serum-free Keratinocyte Growth Medium (KGM, Clonetics, San Diego, CA, USA). All assays were performed on KCs from at least three distinct donors.

Squamous cell carcinoma (SCC) cells were kindly provided by Rheinwald J.G. and cultured following their protocol [31]. For experimental analysis, SCC cells were plated at 3 × 10^4^ cells/well the day before treatment.

For all experiments, HaCaT, primary KCs, and SCC were seeded at densities of 22,000/cm^2^, 33,000/cm^2^, and 16,500/cm^2^, respectively. The viable cell number was evaluated at selected time points following trypsinization and was assessed using a Burker counting chamber by the trypan blue-exclusion method. All the experiments were performed on cells in the logarithmic phase of growth under conditions of > 98% viability.

#### 2.2.2. Nanoparticle Administration

Stock dispersions of CNPs or TNPs were prepared in deionized water at a concentration of 20 mg/mL. Nanoparticles were dispersed with ultrasounds (Branson Ultrasonic corp., Danbury, CT, USA) at 20% amplitude, and immediately diluted to the final concentration in a fresh medium. CNPs and TNPs were added at a final concentration of 200 μg/mL or 100 μg/mL, respectively, after 24 h from cell plating and left overnight prior to cell irradiation.

#### 2.2.3. Cell Irradiation Protocols

HaCaT cells and primary KCs were irradiated 48 h after seeding at room temperature with UV rays according to [23] and at different wavelengths:

UVA (356 nm Spectroline lamp model ENB-260C/FE) (Sigma-Aldrich, St. Louis, MO, USA) at 3 mW/cm^2^ for three times 20 min each separated by two intervals of 10 min each, during which the cells were placed at 37 °C under 5% CO_2_.

UVB (312 nm, Spectroline lamp model ENB-260C/FE) (Sigma-Aldrich, St. Louis, MO, USA) at 3 mW/cm^2^ for 60, 30, 15, and 5 s, which correspond to 0.18, 0.09, 0.045 and 0.015 J/cm^2^, respectively.

#### 2.2.4. Evaluation of Apoptosis

Apoptosis was quantified by visualizing apoptotic nuclei through fluorescence microscopy (ZEISS Axio Observer, ZEISS, Oberkochen, Germany), upon staining with the DNA cell-permeable specific dye Hoechst 33342 (Sigma-Aldrich, St. Louis, MO, USA) directly added into the wells at a final concentration of 10 μg/mL. The fraction of apoptotic nuclei among the total cell population was calculated by counting 300 cells in at least three independent, randomly selected, microscopic fields [32,33].

#### 2.2.5. DNA Damage Analysis by the Alkaline Comet Assay

An alkaline comet assay is a single-cell gel electrophoresis method that allows the detection of both single and double DNA breaks.

The experiment was performed according to protocols described in the literature [34,35]. HaCaT were detached prior to and immediately after irradiation while primary KCs were detached prior, at 1 h and 24 h after irradiation. Cells were then resuspended in 0.5% low melting point agarose and poured onto a glass microscope slide precoated with a layer of 1% normal melting point agarose (NMA), which was necessary to allow for adhesion of the sample. The so-prepared glass slides are left for 10 min on ice in order to allow gel solidification. Slides were then incubated in a lysis solution at pH 10 for 3 h in dark. The lysis mixture was prepared by adding 10% DMSO and 1% Triton X-100 to the lysis solution (2.5 M NaCl, 100 mM EDTA, 10 mM Tris-HCl, and 0.2 mM NaOH in deionized water).

After lysis, slides were rinsed with an alkaline running buffer (pH 13) for 15 min in the dark to allow DNA unwinding. Alkaline comet assay electrophoresis buffer was prepared with 30 mL of 10 mM NaOH and 5 mL of 200 mM EDTA, pH 10, in a total volume of 1 L of deionized water at a temperature of 4 °C.

HaCaT cells and primary KCs electrophoresis were conducted for 25 min using 0.81 V/cm and 300 mA or for 30 min using 1.00 V/cm, respectively, in a unit subcell GT system /15 × 25 cm system equipped with PowerPac™ HC High-Current Power Supply (Bio Rad Laboratories Inc., Hercules, CA, USA). Slides were then gently washed for 5 min in the neutralization buffer solution at pH 7.5 (0.4 M Tris-HCl), washed in deionized water, and left to dry in the dark at room temperature.

Slides were finally stained with 50 μL of ethidium bromide (25 μg/mL) and analyzed by fluorescence microscopy. Images were captured using a ZEISS Axio Observer (ZEISS, Oberkochen, Germany) microscope. HaCaT samples were analyzed using the image processing program ImageJ with the Open Comet (v 1.3) tool plugin. For each sample, 100 comets were analyzed considering the tail DNA score. Primary KC images were analyzed using Carl Zeiss Microscopy GmbH’s ZEN 3.0 software. For each sample, 100 comets have been analyzed and divided into five damage categories (C0–C4), C0 being the least, as previously described [36,37]. The index of damage (ID) was calculated through the equation reported below:


ID=0×n °C0+1×n °C1+2×n °C2+3×n °C3+4×n °C4



n °Cx = number of cells in each category of damage.


All reagents were purchased from Sigma-Aldrich, St. Louis, MO, USA.

#### 2.2.6. Phospho-Histone H_2_AX: Microscope Immunofluorescence

Untreated or irradiated cells were fixed using 4% paraformaldehyde for 15 min. Cells were washed three times with PBS and blocked for 60 min with a blocking buffer solution. Samples were then incubated with a primary antibody against 𝛾-H_2_AX (Cell Signaling, Danvers, MA, USA) in an antibody dilution buffer (PBS, 0.2% Triton X-100 and 1% BSA) overnight at 4 °C. Samples were then rinsed three times with PBS and incubated with a FITC-conjugated secondary antibody (Sigma-Aldrich, St. Louis, MO, USA) for 2 h at room temperature. Samples were washed three times with PBS for 5 min and counterstained with DAPI (2 μg/mL) (Sigma-Aldrich, St. Louis, MO, USA) for 5 min at room temperature. Images were captured using a ZEISS Axio Observer (ZEISS, Oberkochen, Germany) microscope and analyzed using the Carl Zeiss Microscopy GmbH’s ZEN 3.0 software.

#### 2.2.7. Micronucleus Cytome Assay

Micronuclei are small nuclear bodies arising from improper chromosome separation during mitosis as a consequence of the chromosomal loss or unrepaired DNA damage. Evaluation of the number of micronuclei among cells undergoing the mitotic telophase is a measure of early mutagenesis after genotoxic treatments.

Cytochalasin B (3 μg/mL) (Sigma-Aldrich, St. Louis, MO, USA) was added to the cell samples to prevent cytokinesis without inhibiting mitosis before or immediately after UVB irradiation. After 24 and 48 h, the resulting binucleated cells label those that underwent mitosis. After medium removal, cells were rinsed with PBS and treated with a hypotonic solution (0.075 M KCl) for 3 min, then fixed with Carnoy fixative (methanol/acetic acid, 20:1) for 20 min. After washing with PBS, cells were stained with Hoechst 33342 (10 μg/mL) (Sigma-Aldrich, St. Louis, MO, USA) or DAPI 2 μg/mL and with Eosin (Sigma-Aldrich, St. Louis, MO, USA) for 1 min at room temperature. Samples were then rinsed with water and finally resuspended in PBS.

Samples were scored for each experimental point using Axiobserver7 Zeiss fluorescence microscopy (ZEISS, Oberkochen, Germany), following classification criteria of the standard protocol [38]. The number of binucleated cells was evaluated on 1000 counted cells, whereas the number of cells containing micronuclei was evaluated on ≥ 300 binucleated cells.

### 2.3. Statistical Analysis

Each experiment was repeated ≥ 3 times. For data presentation, the mean ± SD was reported. Statistical evaluation was conducted by Student’s *t*-test (significance set at *p* < 0.05), using Past 4.06 b software.

## 3. Results

### 3.1. Nanoceria Reduce UVB-Induced Cytotoxicity in HaCaT Cells

HaCaT cells are human, non-tumor keratinocytes spontaneously immortalized in vitro due to biallelic p53 loss-of-function mutations [39]. They are the reference system for studies on the mechanism of UV-induced cell damage and carcinogenesis.

Nanoceria were synthesized by TEMED-induced precipitation as previously reported [40]. Figure 1A shows the XRD pattern and Miller indexes for the dried powder. SEM observations revealed that the powder consisted of nanometric semispherical particles with sizes in the 7–11 nm range (Figure 1B). A Scherrer analysis on the (111) peak showed a similar size, confirming that the nanoparticles are made of single crystals. The main physicochemical characteristics are reported in Appendix A.

The 200 μg/mL dose was selected as the lowest exerting the maximum cytoprotective effect [41]. Nanoceria were applied 24 h before irradiation and let settle over the cell monolayer.

Cells were irradiated with UVB, the most relevant type of UV rays in terms of public health concern, because they are not completely shielded by the ozone layer therefore reaching earth’s surface, on one side, and recognized as a carcinogenesis risk factor for their direct DNA damaging mechanisms [42,43], on the other.

To select the irradiation exposure dose, we analyzed the effect of 0.18, 0.09, 0.045, and 0.015 J/cm^2^, monitoring proliferation and apoptosis at 24 h post-irradiation.

Data on cell counts are presented as a fold increase with respect to control (posed = 1). As shown in Figure 2A, UVB irradiation decreased HaCaT growth rate according to the exposure dose. Apoptosis data are provided as the fraction of apoptotic cells at 24 h post-exposure (Figure 2B), evaluated on the basis of nuclear vesiculation as described [32,33,44].

Consistently with the reduction in cell number, apoptosis is induced in an exposure-dependent manner, apart from the lower exposure dose that was not effective. Since our goal was to assess whether nanoceria may modulate HaCaT response to UV, we chose for further analyses the exposure dose of 0.045 J/cm^2^, this being the experimental point where apoptosis approached 50%, the parameter that was used as the criterion for choosing the irradiation intensity in our previous study [23].

Figure 2C shows the time course of cytotoxicity parameters in irradiated HaCaT. UVB irradiation induced a strong impairment in the proliferation rate. After an initial drop in cell number, cells slowly recovered, resuming proliferation though at a very slow rate. After seven days, cells reacquired the same proliferation rate as the untreated cells. UVB-induced apoptosis peaked 24 h after treatment (Figure 2D), reaching values of about 60%, slowly decreasing thereafter (Figure 2D).

As expected from our previous models, nanoceria strongly protected against UVB cytotoxicity, reducing the initial cell loss, slowing down proliferation thereafter, and anticipating the resumption of the normal proliferation rate (Figure 2C). Accordingly, nanoceria strongly reduced UVB-induced apoptosis, delaying the peak of cell death from 24 to 48 h post-irradiation, and reducing its extent at all the time points tested, though without changing the kinetics of apoptosis (Figure 2D).

### 3.2. Nanoceria Reduce DNA Damage and Mutagenesis in HaCaT Cells

We then explored UV-induced DNA damage by an alkaline comet assay; the results are presented as DNA damage index for each treatment. Analyses were performed immediately after irradiation, to limit as much as possible that DNA repair altered the initial break load.

As expected, UVB caused DNA damage, inducing breaks with a fourfold increase with respect to the control (Figure 3A). Strikingly, nanoceria were highly protective, almost halving UV-induced DNA damage. Notably, nanoceria also reduced the basal comet signal (Figure 3A).

These results show that the cytoprotective effect of nanoceria correlates with a decrease in UVB-induced DNA damage.

Since UVB irradiation induces DNA lesions, abnormal anaphases might occur at the following mitosis, leading to the formation of micronuclei, which are considered a reliable index of general mutagenesis. To assess the mutagenic rate, we performed a micronucleus cytome assay at 24 h and 48 h post-treatment.

As expected, irradiation induced micronuclei formation after 24 h, which were strongly reduced (almost halved) by nanoceria (Figure 3B). At 48 h post-irradiation micronuclei decreased, possibly because mutant cells underwent apoptosis; however, nanoceria protection remained evident also at this time-point (Figure 3B).

These results show that nanoceria exert a protective action also against the consequences of UVB-induced DNA damage, possibly including carcinogenesis.

### 3.3. Nanotitania Increase UVA-Cytotoxicity, Which Is Prevented by Nanoceria

Nanotitania are such excellent shields that the pro-oxidant photocatalytic effect is often neglected, since most areas of the skin are protected from nanoparticle intrusion by the stratum corneum. However, in the presence of skin discontinuities, nanotitania may physically interact with living cells, and become toxic upon irradiation. We explored the effects of irradiated nanotitania in physical contact with living cells, showing that they exert a strong cytotoxic and genotoxic effect [23]. This is especially evident upon UVA irradiation, which, in the experimental conditions, exerted poor toxicity of their own (whereas UVB and UVC effects were too strong to allow appreciation of the extra toxicity). Notably, in that study, entirely focused on the extra-shielding nanoparticle effects, nanotitania could not display their UV-screen power since Jurkat cells grow in suspension. Here, instead, we used monolayer HaCaT cells, and in this system, nanotitania can intercept UV photons since it is administered above a cell monolayer. This allows mimicking an irradiated compromised skin devoid of a stratum corneum, where both nanotitania effects (shielding vs. pro-oxidant) are simultaneously present.

Nanotitania were synthesized according to the previously described wet chemical synthetic procedure [23]. The main physico-chemical features of nanotitania are reported in Appendix A. Figure 4A shows the XRD diffraction pattern of TNPs.

As shown in Figure 4B, in the presence of nanotitania, the levels of UVA-induced apoptosis are almost doubled (1.8 fold), indicating that the cytotoxic effect of irradiated nanotitania overcomes their shielding effect.

Instead, nanoceria strongly protected from UVA-induced apoptosis (Figure 4B). Strikingly, nanoceria were also able to counteract cytotoxicity induced by irradiated nanotitania, dose-dependently relieving their noxious effects in a mixed-nanoparticle preparation (Figure 4B). Nanoceria were so effective that, when nanoparticles were present in a ratio of 1:1, the extent of apoptosis dropped below control values.

### 3.4. Nanoceria Reduce UVB-Induced Cell Loss in Primary Human Keratinocytes

To confirm the above-reported data in primary cells, we repeated key experiments on human primary keratinocytes, using the same exposure conditions as for HaCaT cells.

As shown in Figure 5A, UVB irradiation caused a sudden stop of proliferation in the primary keratinocytes of all three donors, though with different kinetics (Figure 5A). This is due to a halt in cell cycle progression, as shown by the fraction of cells capable of duplication, measured in the presence of the cytokinesis inhibitor cytochalasin B, which allows for visualizing proliferating cells as binucleated (Figure 5B).

Micrographs of control and irradiated keratinocytes are shown in Figure 5C. The reduction in cell number induced by UVB-induced in the latter is evident, and is associated with cell enlargement (see red asterisks), as often occurs upon DNA damage [45]. Morphologies of the three donors’ cells were very much similar to each other, both in control and irradiated cells (data not shown). Also in primary keratinocytes, the presence of nanoceria deeply contrasted with the antiproliferative effect of UVB, reducing cell loss and repopulation slow-down (Figure 5A), and increasing the fraction of bi-nucleated cells in the presence of the cytokinesis inhibitor cytochalasin B (Figure 5B). This was confirmed by the microscopic morphological analysis, showing that nanoceria allows irradiated cells to form a denser monolayer, without cell enlargement, maintaining a morphology similar to non-irradiated cells (Figure 5C).

In parallel, the analysis of apoptosis performed as the fraction of cells with vesiculated nuclei [32,33,44], showed that UVB induced apoptosis, to an extent ranging from 20 to 50%, and with a peak between one to seven days post-irradiation, according to the individual donors (Figure 5D). Notably, nanoceria protect from apoptosis in all cases (Figure 5D).

### 3.5. Nanoceria Reduce DNA Damage and Prevent Mutagenesis in Primary Human Keratinocytes

We explored the mechanism of nanoceria protection from UVB irradiation, focusing on DNA damage and its genotoxic consequences, being mutagenesis induced by solar exposure a major factor determining skin carcinogenesis.

To directly measure DNA damage in normal human keratinocytes, we performed an alkaline comet assay on keratinocytes irradiated ± nanoceria, showing heavy damage at 1 h post-irradiation, which was strongly blunted by nanoceria (Figure 6A).

γ-H_2_AX is a hallmark of cell response to DNA breaks and indicates that cells are actively repairing the damage [40]. We therefore analysed the γ-H_2_AX signal, showing it is activated 1 h post-irradiation, and shut off at 24 h. This indicates full DNA repair, confirming what shown above for the comet assay (Figure 6B).

Inefficient DNA repair is a major cause of mutagenesis and, in the long run, of cancer. UVB are recognized as the main inducers of skin cancer via DNA damage [46]. Here, we show that UVB-promoted mutagenesis (threefold over the control) is not only reduced, but almost totally prevented, by nanoceria (Figure 6C).

### 3.6. Nanoceria Favour Basal Proliferation of Primary Human Keratinocytes but Not of HaCaT or Squamous Cell Carcinoma Keratinocytes

A striking difference between primary and immortalized keratinocytes relies on the effect of nanoceria on the basal proliferation rate (i.e., without irradiation), which was substantially boosted only in the former. Notably, the basal proliferation of p53-null keratinocyte cell lines derived from squamous cell carcinoma (SCC) are also unaffected by nanoceria (Figure 7).

## 4. Discussion

In this study, we showed that nanoceria exert a strong antioxidant-based effect, protecting keratinocytes from the cytotoxic and genotoxic effects of UV irradiation. Nanoceria not only were able to reduce apoptosis and help irradiated cells resume the basal proliferation rate, but were also capable of reducing UVB-induced DNA damage and mutagenesis. This latter effect is especially important considering that solar exposure is the major cause of the increase in skin cancer in the last century [46].

Our data, obtained in HaCaT cells and confirmed in primary human keratinocytes, complement what has been reported in recent studies on the effect of nanoceria on UV-irradiated normal fibroblasts [47,48], and substantially corroborate the message of our previous study performed on the lymphocyte cell line Jurkat [23]. In particular, with the present work, we transferred the knowledge acquired from the experiments performed on a reference cell line (Jurkat cells, a historical system of choice where UVB genotoxic effects were studied [49]) to a major real target of UV damage, i.e., keratinocytes, over which sunscreen formulations are actually applied. This constitutes a key passage providing proof-of-principle evidence for the possible successful inclusion of nanoceria in commercial cosmetic solar shields.

HaCaT cells are a very popular experimental system to study UV-induced effects in vitro. However, they are immortalized, and if this assures a stabilized and reliable system, it also implies reduced tumor suppressor activities and different responses to DNA-damaging treatments. In particular, HaCaT underwent spontaneous in vitro mutation in p53, which is present but completely nonfunctional [39]. Since UV induce DNA damage, this p53 deficiency created a deep concern on our side, which pushed us to confirm the results obtained with HaCaT in primary human keratinocytes, a more heterogeneous (and troublesome) experimental system. Nevertheless, experiments with nanoceria on HaCaT cells vs. primary keratinocytes gave substantially similar results, at least as far as the UV-protective effect is concerned.

Instead, a different effect was found in non-irradiated HaCaT vs. primary keratinocytes. The finding that nanoceria favour the proliferation of normal keratinocytes fits with the notion that nanoceria topical administration facilitates the healing of dermal wounds in mice [50], since sub-confluence in a culture flask may resemble tissues needing to be replenished. Notably, we report that neither HaCaT nor squamous cell carcinoma cell lines proliferation is affected by nanoceria (see Figure 7). We have very similar evidence on a pair of normal vs. p53-null cancer prostate cells, where nanoceria promote the proliferation of the former while not affecting the latter [Corsi et al., work in progress]. This highlights a further different effect of nanoceria in p53 proficient vs. deficient cells, whereas the oncogenic asset, normal in HaCaT though mutant in SCC or the prostate cancer cells, seems here less relevant. These findings indicate that the ability of nanoceria to speed up wound healing, being possibly limited to normal cells, may help wounded tissues favouring replication of p53^wt^ cells during wound repair. Interestingly, recent studies by Martincorena et al., have shown that skin tissues from elderly people, though histologically normal, contain frequent clones of cells mutant in p53 [51]. It is tempting to speculate that nanoceria may help rejuvenating such tissues when wounded.

Cosmeceutical sunscreens are very efficient in protecting exposed skin from the immediate noxious effects of solar exposure, such as, e.g., sunburns and erythema. However, epidemiology studies point out that protection hardly applies to the long-term effects of exposure, such as carcinogenesis [25,26]. This calls for more efficient protection.

Modern sunscreen lotions based on inorganic nano-formulated active principles are highly durable since the high-tech materials are able to maintain the initial UV-absorption ability, unlike the traditional organic filters mimicking our endogenous melanin [52].

Our results on the toxicity of irradiated nanotitania reinforce the concern of a possible hazard of nanotitania if irradiated on compromised skin. This is in line with other reports, showing that nanotitania are toxic when irradiated in contact with cells, though not if irradiated in 3D models of skin equivalent possessing a pseudo-stratum corneum [53]. This highlights the increasing urgency to find reliable tools that improves commonly marketed sunscreens, mitigating the adverse effect related to irradiated nanotitania. Other commonly used inorganic filters, such as e.g., zinc oxide nanoparticles, are also subject to photocatalysis, thus being not an adequate alternative [54].

To prevent this health hazard, the inclusion of antioxidants into nanoparticle-based sunscreens was often attempted, using carotenoids, vitamins (C and E), and plant extracts; however, these organic compounds have the main limitation of short-term duration due to lack of recycling ability in an abiotic environment such as outer skin. The consequence is that their topical use requires too frequent re-application, being, in fact, a non-sensible option [55,56,57]. To overcome this problem, in a recent work, it has been proposed to add biodegradable nanoparticles presenting both durable superoxide dismutase and catalase enzyme activity [58]. Nanoceria, which also possess combined superoxide dismutase and catalase activity, have the exceptional bonus that such enzymatic activities do not require a biological environment that provides energy to allow recycling. Indeed, the nanoceria redox cycle is self-maintained, being fueled by superoxide and peroxide that cyclically oxidized and reduce Ce atoms, in an energy-free self-regenerating way, up to the point that a single application can protect cells for > 10 days [59].

Considering that nanoceria also possess a redox-independent UV-shielding ability, it seems reasonable, and indeed it was proposed, to use them as double-acting agents in sunscreens as an alternative to those currently used [60]. However, it is generally reported, even with some exceptions (e.g., [60]), that nanoceria are less proficient UV shields with respect to nanotitania, as we also observed by comparing two nanoparticle preparations with the same size [23]. This makes this strategy less appealing, because if it is important to consider safety measures for eventual skin abrasions, nonetheless, the larger proportion of exposed skin is intact, and its protection remains the first objective of the solar filters. Hence, the presence of potent shields should be a priority. Our findings that nanoceria scavenge ROS produced by the photocatalytic effect of irradiated nanotitania, and that this is functionally translated into the prevention of irradiated nanotitania cytotoxicity, indicate a nanotitania-nanoceria mixed-nanoparticle preparation as a potentially successful strategy. In this mix, nanotitania would provide most of the UV shielding effect, whereas nanoceria, though further supporting the shielding function, would primarily act as scavengers for the ROS produced by the nanotitania photocatalytic effect, thus protecting both intact and damaged skin. This points to a possible breakthrough commercial product including nanoceria in nanotitania- (or zinc dioxide-) based formulations, which may not only help reducing the immediate noxious effects of UV, but also those that appear in the long run, and that are definitely more serious, such as chronic inflammation and carcinogenesis.

## Figures and Tables

**Figure 1 antioxidants-12-00757-f001:**
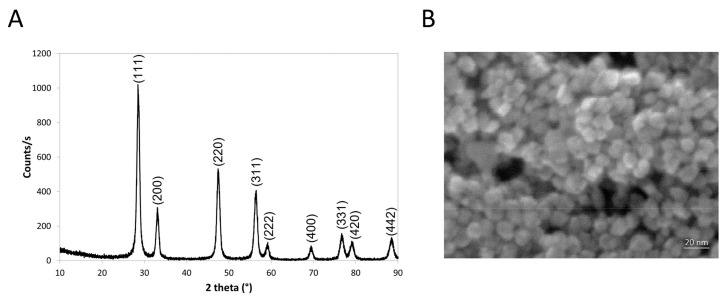
Nanoceria physicochemical characterization. (**A**) XRD diffraction patterns and Miller indexes of nanoceria. (**B**) Scanning electron microscopy (SEM) image of nanoceria. The scale bar corresponds to 20 nm.

**Figure 2 antioxidants-12-00757-f002:**
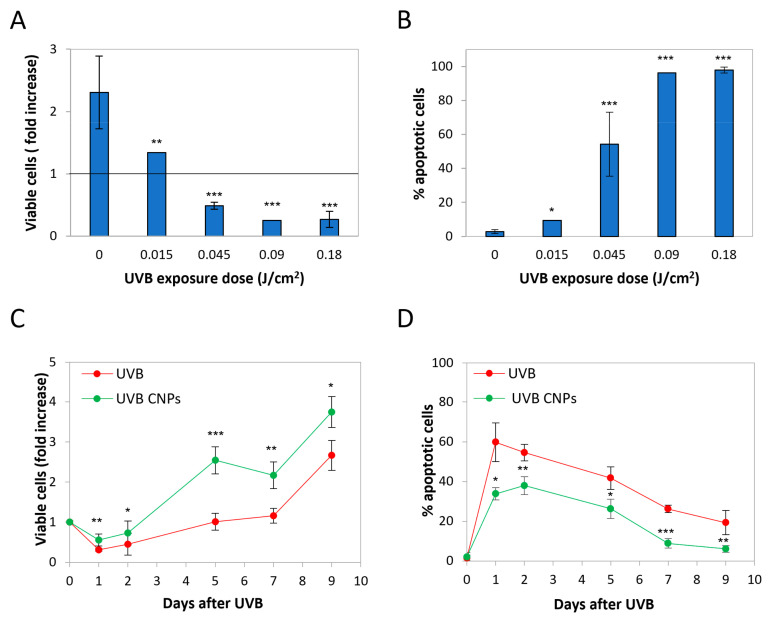
Nanoceria reduced UVB-induced cytotoxicity in HaCaT cells. (**A**) Cell number and (**B**) apoptosis of HaCaT cells at 24 h following UVB 0.18 J/cm^2^ exposure. The viable cell ratio was obtained as a normalized value over day 0. Statistical significance was calculated via Student’s *t*-test. * *p* < 0.05, ** *p* < 0.01 and *** *p* < 0.001, with respect to UVB-irradiated cells. (**C**) Time course of viable (trypan blue-excluding) cell number (expressed as fold over day 0) upon 15 s exposure to UVB ± nanoceria (CNPs). (**D**) Kinetics of the fraction of apoptotic cells upon 0.18 J/cm^2^ exposure to UVB ± nanoceria (CNPs). Statistical significance was calculated via Student’s *t*-test. * *p* < 0.05, ** *p* < 0.01, *** *p* < 0.001, with respect to UVB-irradiated cells.

**Figure 3 antioxidants-12-00757-f003:**
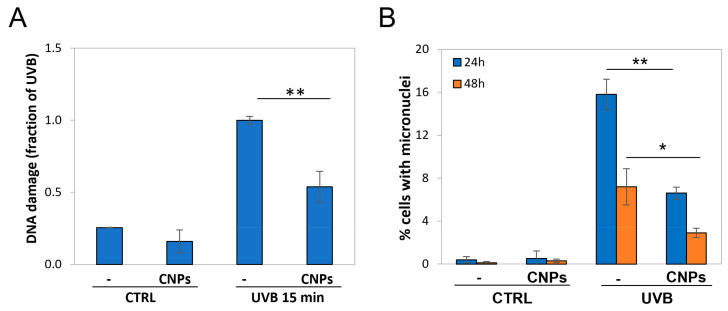
Nanoceria reduced DNA damage and prevented mutagenesis in HaCaT cells. (**A**) Comet assay analysis of UVB-irradiated HaCaT ± nanoceria (CNPs). Values are normalized with respect to the maximum value (UVB 1h) posed = 1. Statistical significance was calculated via Student’s *t*-test. ** *p* < 0.01, with respect to UVB-irradiated cells. (**B**) Micronuclei after UVB exposure ± nanoceria (CNPs). Nanoceria pretreatment halves UVB-induced micronuclei. Statistical significance was calculated via Student’s *t*-test. * *p* < 0.05 and ** *p* < 0.001 with respect to UVB-irradiated cells.

**Figure 4 antioxidants-12-00757-f004:**
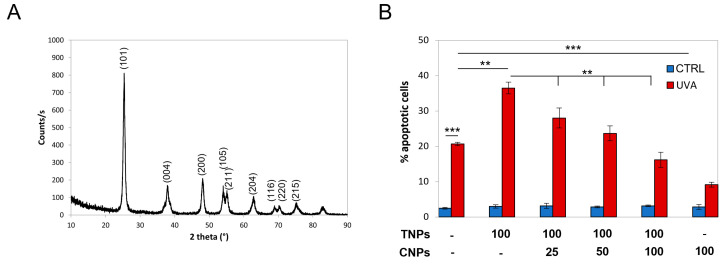
Nanotitania increases UVA-cytotoxicity, which is prevented by nanoceria. (**A**) XRD diffraction pattern and Miller indexes of nanotitania (TNPs). (**B**) Nanoceria (CNPs) dose-dependently reduce apoptosis induced by UVA-irradiated TNPs-induced apoptosis. HaCaT cells were exposed to UVA ± nanoparticles as described (concentration indicates μg/mL). The fraction of apoptotic cells is quantified by nuclear morphology as described. Statistical significance was calculated via Student’s *t*-test. ** *p* < 0.01 and *** *p* < 0.001.

**Figure 5 antioxidants-12-00757-f005:**
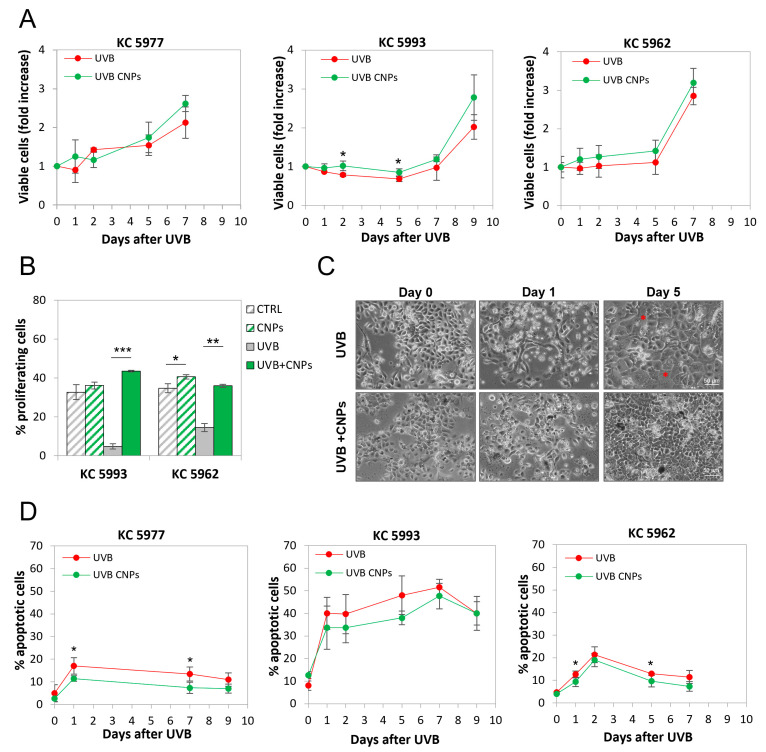
Nanoceria reduces UVB-induced cell loss in primary human keratinocytes from healthy donors. Keratinocytes (KC) from healthy donors (5977, 5993, 5962) were exposed to UVB ± nanoceria (CNPs) for 15 s and let recover for 7–9 days. (**A**) Kinetics of viable (trypan blue-excluding) cell number expressed as fold over day 0. Statistical significance was calculated via Student’s *t*-test. * *p* < 0.05, with respect to irradiated cells. (**B**) Nanoceria (CNPs) reduce UV-induced keratinocyte cytostasis. The fraction of binucleated cells at 48 h after UVB exposure in the presence of the cytokinase inhibitor cytochalasin B is shown. Statistical significance was calculated via Student’s *t*-test. * *p* < 0.05, ** *p* < 0.01 and *** *p* < 0.001. (**C**) Phase-contrast images of UVB-irradiated cells ± nanoceria (CNPs) at the indicated time points. A * indicates DNA damage-induced enlarged cells. (**D**) Time course of apoptotic cells after UVB exposure ± nanoceria (CNPs). Statistical significance was calculated via Student’s *t*-test. * *p* < 0.05 with respect to irradiated cells.

**Figure 6 antioxidants-12-00757-f006:**
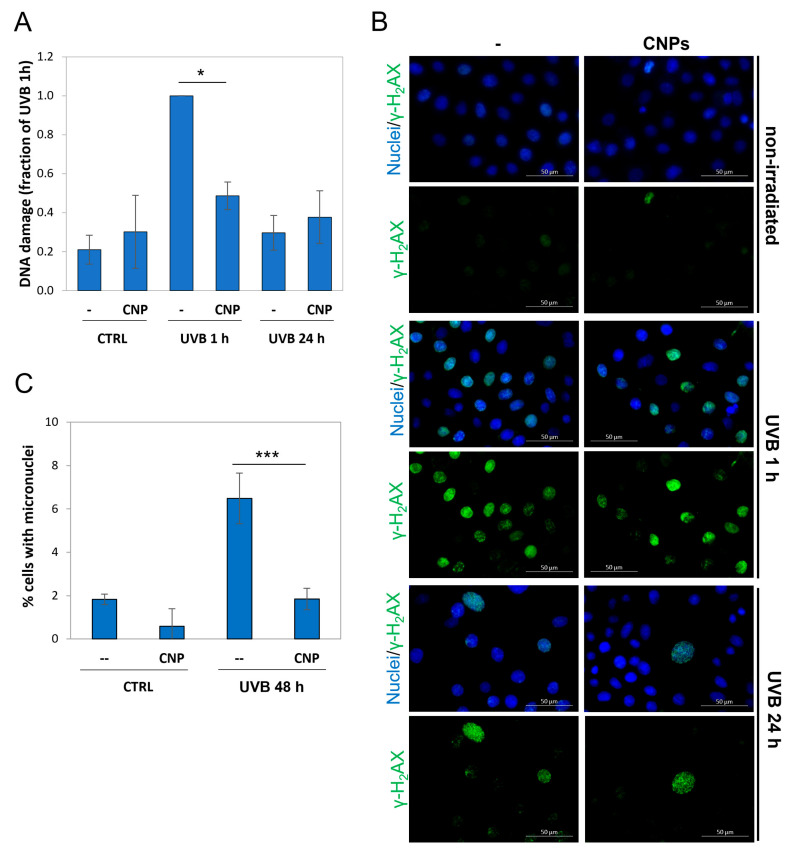
Nanoceria reduces DNA damage and prevent mutagenesis in primary human keratinocytes. (**A**) Comet assay of primary keratinocytes (KC 5993) ± nanoceria (CNPs) at 1 h or 24 h after UVB exposure. Values are normalized with respect to the maximum value (UVB 1h) posed = 1. Statistical significance was calculated via Student’s *t*-test. * *p* < 0.05. (**B**) Immunofluorescence with anti-γ-H_2_AX antibody and DAPI staining of KC 5993 ± nanoceria (CNPs) at 1 h or 24 h after UVB exposure in wide-field micrography. The scale bar corresponds to 50 μm. (**C**) Percentage of cells containing micronuclei among binucleated KC 5993 cells 48 h after UVB irradiation ± nanoceria (CNPs). Statistical significance was calculated via Student’s *t*-test. *** *p* < 0.001.

**Figure 7 antioxidants-12-00757-f007:**
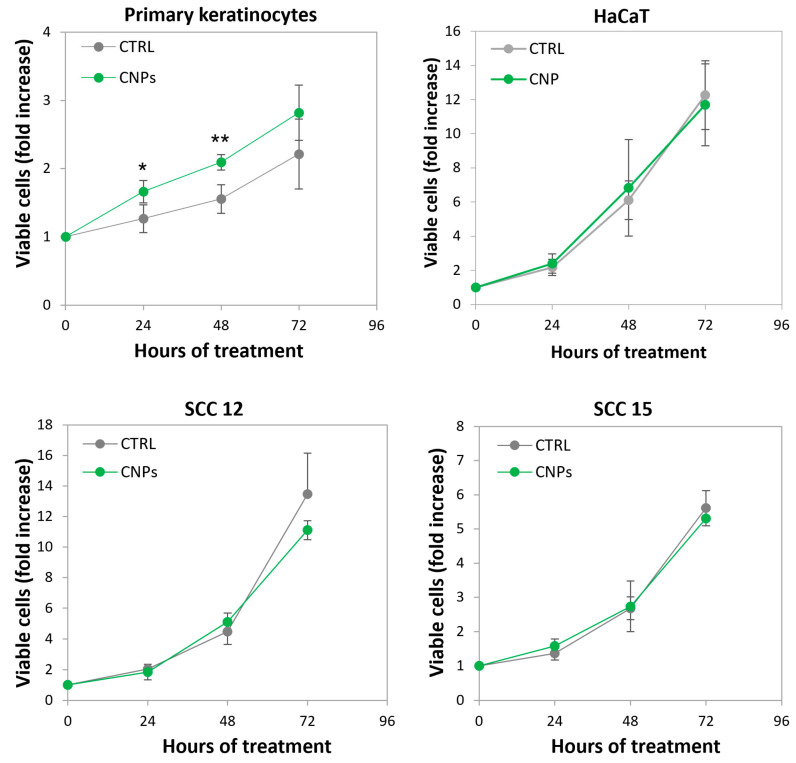
Nanoceria favour the proliferation of normal keratinocytes but not of HaCaT or squamous cell carcinoma keratinocytes. Nanoceria (CNPs) were administered at day 0 to primary keratinocytes, HaCaT, and two squamous cell carcinoma keratinocytes cell lines (SCC 12 and SCC 15). Kinetics of viable (trypan blue excluding) cell number expressed as fold over day 0. Statistical significance was calculated via Student’s *t*-test. * *p* < 0.05, ** *p* < 0.01 with respect to untreated cells.

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
