# Peer review of "Safe-Shields: Basal and Anti-UV Protection of Human Keratinocytes by Redox-Active Cerium Oxide Nanoparticles Prevents UVB-Induced Mutagenesis"

_antioxidants, 2023, doi:10.3390/antiox12030757_

Round 1

Reviewer 1 Report

The paper reports on the study of the UV-protective properties of antioxidant cerium oxide NPs. On an example of primary human keratinocytes, the ability of CeO2 NPs to protect the cells from UV-induced DNA damage is demonstrated. The subject of the paper falls within the scope of Antioxidants journal. The results obtained are new and worth of being published.

I have the following comments:

1. In the Introduction section, it is mentioned that nano-titania changes their conformation while absorbing UV-photons. This seems to be incorrect, since electronic excitation cannot be considered as a conformation change.

2. Please correct the chemical formula of cerium(III) nitrate (line 105).

3. Information on the particle size distribution and dzeta potential of CeO2 NPs suspensions in deionized water should be provided.

4. It is still unclear what is the mechanism of the protective action of CeO2 NPs? Were they internalized by keratinocytes, or no?

5. I would suggest including some additional references concerning UV-protective action of nano-ceria. See, e.g. 10.3390/antiox12010190 and the Introduction to this paper.

Author Response

REFEREE 1

The paper reports on the study of the UV-protective properties of antioxidant cerium oxide NPs. On an example of primary human keratinocytes, the ability of CeO2 NPs to protect the cells from UV-induced DNA damage is demonstrated. The subject of the paper falls within the scope of Antioxidants journal. The results obtained are new and worth of being published.

I have the following comments:

  1. In the Introduction section, it is mentioned that nano-titania changes their conformation while absorbing UV-photons. This seems to be incorrect since electronic excitation cannot be considered as a conformation change.

Right, thanks for pointing out. We re-formulated the all period that now runs: ”Especially in the anatase form, nanotitania are highly efficient UV-shields absorbing UV-photons [10.1111/phpp.12214], thus providing long-lasting protection [17]. This implies production of ROS through a process known as photocatalytic effect, which allows dissipating the extra-energy deriving form UV-photon absorption [18].”

  1. Please correct the chemical formula of cerium(III) nitrate (line 105).

Sorry; we fixed the mistake.

  1. Information on the particle size distribution and dzeta potential of CeO2 NPs suspensions in deionized water should be provided.

We added the information as requested; please see Supplementary Materials.  

  1. It is still unclear what is the mechanism of the protective action of CeO2 NPs? Were they internalized by keratinocytes, or no?

NP internalization, and their role in the bio-phenomena, is a key and complex point of the research on NP bioeffects. Showing internalization does not show that the effects occur inside the cells; vice-versa, lack of internalization does not exclude important NP bioactivity. For example, we know that Jurkat cells, where nanoceria exert potent protection against UV, only poorly internalize NP, and we are working on the mechanism at the basis of this “action from outside”. The issue is very complex and ill-defined; we are still working on its basic principles, and do not feel ready yet to publicly comment the issue.

  1. I would suggest including some additional references concerning UV-protective action of nano-ceria. See, e.g. 10.3390/antiox12010190 and the Introduction to this paper.

As suggested by the Reviewer, we added reference of the recent papers showing nanoceria effects on UV-irradiated cells.

Reviewer 2 Report

The manuscript reports on UV-protective properties of cerium oxide, which prevents UVB-induced mutagenesis of human keratinocytes. The study also includes the assessment of UVA-cytotoxicity of titanium dioxide. According to the paper, nanoceria reduce UVB-induced cytotoxicity in cell lines, HaCaT cells and squamous cell carcinoma. An essential advantage of the work is examination of UVB-induced cell loss on primary human keranocytes. The data on the cytoprotective properties of nanoceria allow considering it as a promising component of sunscreens for reducing the cytotoxicity of UV-protective nanotitania. The results obtained are of special interest to the readership of the Antioxidants journal. The paper is written in good English, the experimental results are solid and presented in a logical manner. The manuscript is suitable for publication in the Antioxidants journal with some minor corrections.

  1. Please provide more details on the characterization of cerium dioxide and titanium dioxide (XRD, SEM, etc.), e.g. in Supplementary materials.
  2. Please estimate the size of ceria and titania NPs from XRD and low temperature nitrogen adsorption data.
  3. Please, provide some data on the stability and particle size (I mean hydrodynamic diameter and zeta potential) of the aqueous dispersions of CNPs and TNPs.
  4. The choice of a relevant system for evaluating UV-induced changes in biological systems is quite important. In this regard, please provide a short (maybe) discussion of enzyme-like activity of ceria in biologically relevant media of different chemical composition. I mean the species that can affect the activity of CeO2, such as phosphate ion or proteins. The acidity of the environment should also be mentioned in this context.
  5. Please add some discussion (or experimental data if applicaple) for the internalization of nanoceria by the cells.

Author Response

REFEREE 2

The manuscript reports on UV-protective properties of cerium oxide, which prevents UVB-induced mutagenesis of human keratinocytes. The study also includes the assessment of UVA-cytotoxicity of titanium dioxide. According to the paper, nanoceria reduce UVB-induced cytotoxicity in cell lines, HaCaT cells and squamous cell carcinoma. An essential advantage of the work is examination of UVB-induced cell loss on primary human keratinocytes. The data on the cytoprotective properties of nanoceria allow considering it as a promising component of sunscreens for reducing the cytotoxicity of UV-protective nanotitania. The results obtained are of special interest to the readership of the Antioxidants journal. The paper is written in good English, the experimental results are solid and presented in a logical manner. The manuscript is suitable for publication in the Antioxidants journal with some minor corrections.

1) Please provide more details on the characterization of cerium dioxide and titanium dioxide (XRD, SEM, etc.), e.g. in Supplementary materials.

2) Please estimate the size of ceria and titania NPs from XRD and low temperature nitrogen adsorption data.

3) Please, provide some data on the stability and particle size (I mean hydrodynamic diameter and zeta potential) of the aqueous dispersions of CNPs and TNPs.

We group the response to the three points. In particular, we provided to add in the manuscript the following required information:

  1. ) XRD and SEM analyses of CNPs (Figure 1);
  2. ) XRD pattern of TNPs (Figure 4);
  3. ) Crystal structure, size of the nanoparticles (estimated by SEM), single point specific surface area (measured by BET), mean hydrodynamic diameter and zeta potential in deionized water (Supplementary Materials).

We also estimated the size of the crystallite from XRD patterns by measuring the full-width half-maximum (FWHM) of the characteristic peak using the Scherrer equation, obtaining similar size to SEM measurements (see paragraph 3.1).

4) The choice of a relevant system for evaluating UV-induced changes in biological systems is quite important. In this regard, please provide a short (maybe) discussion of enzyme-like activity of ceria in biologically relevant media of different chemical composition. I mean the species that can affect the activity of CeO2, such as phosphate ion or proteins. The acidity of the environment should also be mentioned in this context.

Yes, this is a key point, thanks. We added the following in the Introduction: “To note, nanoceria catalytic activities may be influenced by specific chemical features of biological environment, such as ionic composition or acidity [https://doi.org/10.1016/j.biomaterials.2011.05.073]. For example, strongly acidic pH (> 4) differently affect the catalase vs. SOD-mimetic activities, leading to peroxide accumulation in acidic (e.g., lysosomic) bio-environment [10.1002/smll.200700824][10.3389/fonc.2018.00309]”

5) Please add some discussion (or experimental data if applicable) for the internalization of nanoceria by the cells.

NP internalization, and their role in the bio-phenomena, is a key and complex point of the research on NP bioeffects. Showing internalization does not show that the effects occur inside the cells; vice-versa, lack of internalization does not exclude important NP bioactivity. For example, we know that Jurkat cells, where nanoceria exert potent protection against UV, only poorly internalize NP, and we are working on the mechanism at the basis of this “action from outside”. The issue is very complex and ill-defined; we are still working on its basic principles, and do not feel ready yet to publicly comment the issues.

Reviewer 3 Report

The manuscript titled "Safe-shields: basal and anti-UV protection of human keratinocytes by redox-active cerium oxide nanoparticles prevents UVB-induced mutagenesis" presents findings that demonstrate the protective effects of nanoceria on human keratinocytes. The authors report that nanoceria promote the basal proliferation of primary normal keratinocytes while also protecting them from UVB-induced DNA damage, mutagenesis, and apoptosis. This results in a reduction of cell loss and acceleration of recovery with no noticeable cellular damage. Similar effects were observed in non-cancerous, but immortalized, p53-null HaCaT keratinocytes. However, two exceptions were noted: nanoceria did not accelerate basal HaCaT proliferation, and upon UVB exposure, nanoceria decoupled DNA damage (detected as Comet) from the response to damage (detected as γ-H2AX). Additionally, nanoceria were found to protect HaCaT cells from oxidative stress induced by irradiated titanium dioxide nanoparticles, a major component of commercial UV-shielding lotions, thereby neutralizing their most critical side effects.

Overall, the manuscript is well-written and fits the scope of the journal and special issue. However, there are two minor remarks. Firstly, it would be beneficial to clearly indicate what is new in this article and how it builds upon reference 19. Secondly, Figure 1B and Figure 4A should be incorporated into the text or prepared as a table, as they are not true figures.

Author Response

REFEREE 3

The manuscript titled "Safe-shields: basal and anti-UV protection of human keratinocytes by redox-active cerium oxide nanoparticles prevents UVB-induced mutagenesis" presents findings that demonstrate the protective effects of nanoceria on human keratinocytes. The authors report that nanoceria promote the basal proliferation of primary normal keratinocytes while also protecting them from UVB-induced DNA damage, mutagenesis, and apoptosis. This results in a reduction of cell loss and acceleration of recovery with no noticeable cellular damage. Similar effects were observed in non-cancerous, but immortalized, p53-null HaCaT keratinocytes. However, two exceptions were noted: nanoceria did not accelerate basal HaCaT proliferation, and upon UVB exposure, nanoceria decoupled DNA damage (detected as Comet) from the response to damage (detected as γ-H2AX). Additionally, nanoceria were found to protect HaCaT cells from oxidative stress induced by irradiated titanium dioxide nanoparticles, a major component of commercial UV-shielding lotions, thereby neutralizing their most critical side effects.

Overall, the manuscript is well-written and fits the scope of the journal and special issue. However, there are two minor remarks.

Firstly, it would be beneficial to clearly indicate what is new in this article and how it builds upon reference 19.

In this revised version, we specified the increment of knowledge of the present paper with respect to our previous study (former ref. n. 19) as follows: “In particular, with the present work, we transfer the knowledge acquired from the experiments performed on a reference cell line (Jurkat cells are historically a system of choice where UVB genotoxic effects were studied [10.1667/RR0991.1]) to a major real target of UV damage, i.e., keratinocytes, over which sunscreen formulations are actually applied. This constitutes a key passage providing proof-of-principle evidences for possible successful inclusion of nanoceria in commercial cosmetic solar shields.”

Secondly, Figure 1B and Figure 4A should be incorporated into the text or prepared as a table, as they are not true figures.

Yes, we agree and moved the data to Supplementary Tables 1 and 2.

Round 2

Reviewer 1 Report

The authors have addressed all of my comments. The paper is now suitable for publication in Antioxidants.